# Primary Generalized Glucocorticoid Resistance and Hypersensitivity Syndromes: A 2021 Update

**DOI:** 10.3390/ijms221910839

**Published:** 2021-10-07

**Authors:** Nicolas C. Nicolaides, Evangelia Charmandari

**Affiliations:** 1First Department of Pediatrics, Division of Endocrinology, Metabolism and Diabetes, “Aghia Sophia” Children’s Hospital, National and Kapodistrian University of Athens Medical School, 11527 Athens, Greece; evangelia.charmandari@googlemail.com; 2Center of Clinical, Experimental Surgery and Translational Research, Division of Endocrinology and Metabolism, Biomedical Research Foundation of the Academy of Athens, 11527 Athens, Greece; 3University Research Institute of Maternal and Child Health and Precision Medicine, University of Athens, 11527 Athens, Greece; 4Department of Molecular Genetics, Function and Therapy, The Cyprus Institute of Neurology and Genetics, Nicosia 2371, Cyprus

**Keywords:** glucocorticoid hypersensitivity, glucocorticoid receptor, glucocorticoid resistance, glucocorticoids, *NR3C1* gene mutations

## Abstract

Glucocorticoids are the final products of the neuroendocrine hypothalamic–pituitary—adrenal axis, and play an important role in the stress response to re-establish homeostasis when it is threatened, or perceived as threatened. These steroid hormones have pleiotropic actions through binding to their cognate receptor, the human glucocorticoid receptor, which functions as a ligand-bound transcription factor inducing or repressing the expression of a large number of target genes. To achieve homeostasis, glucocorticoid signaling should have an optimal effect on all tissues. Indeed, any inappropriate glucocorticoid effect in terms of quantity or quality has been associated with pathologic conditions, which are characterized by short-term or long-lasting detrimental effects. Two such conditions, the primary generalized glucocorticoid resistance and hypersensitivity syndromes, are discussed in this review article. Undoubtedly, the tremendous progress of structural, molecular, and cellular biology, in association with the continued progress of biotechnology, has led to a better and more in-depth understanding of these rare endocrinologic conditions, as well as more effective therapeutic management.

## 1. Glucocorticoid Signaling

Glucocorticoids are steroid hormones, which are biosynthesized in the zona fasciculata of the adrenal cortex, as well as in many other extra-adrenal organs, including the intestine, thymus, and skin [1,2]. These lipophilic molecules are secreted inthe peripheral circulation in response to stressors, as well as in an ultradian and circadian fashion [3,4,5,6], and play a fundamental role in the establishment of resting and threatened homeostasis [7]. Indeed, glucocorticoids act as the end-products of a neuroendocrine axis, the hypothalamic–pituitary–adrenal (HPA) axis. This influences a broad spectrum of physiologic functions, such as carbohydrate, fat, and protein metabolism; cardiovascular tone, memory, behavior, and cognition; as well as inflammatory and immune responses [8,9,10]. In addition to their essential homeostatic functions, glucocorticoids can change the percentage of methylation in many cytosine–guanine dinucleotides (CpG) within a growing number of promoters by altering the expression and/or the activity of methyltransferases or demethylases, ultimately leading to epigenetic modifications of gene expression [11,12]. All these pleiotropic glucocorticoid effects are mediated by genomic, non-genomic, and mitochondrial actions of a ubiquitously expressed protein, the human glucocorticoid receptor (hGR), which is a member of the steroid receptor family of transcription factors [13].

The hGR is expressed by the *NR3C1* gene, which is located on chromosome 5 (5q31.3), and contains 10 exons. Exon 1 is a non-expressing DNA region, and contains multiple regulatory sites that influence the expression of the *NR3C1* gene positively or negatively. Exons 2-9α/9β contribute to the expression of hGR [14]. The alternative usage of exons 9α or 9β produces the two main protein isoforms: the hGRα and the hGRβ. The hGRα is expressed in almost every cell, and is predominantly located in the cytoplasmic region of the target cell. It is capable of binding natural glucocorticoids (cortisol in humans, corticosterone in rodents) and synthetic glucocorticoid analogues, and mediates all the above-mentioned glucocorticoid actions [15]. On the other hand, the hGRβ was initially reported as an inhibitor of the transcriptional activity of hGRα via several molecular mechanisms [16,17,18,19]. Accumulating evidence indicates that this isoform is substantially involved in insulin signal transduction, pathways of gluconeogenesis, and inflammatory processes in mouse liver [19,20,21]. Furthermore, the hGRβ was found to contribute to the formation of gliomas and migration of bladder malignant cells, suggesting an evolving role of this receptor isoform in crucial steps of carcinogenesis [19,22,23,24]. In addition to these two protein isoforms, Lu and Cidlowski proved the existence of eight additional GRα isoforms (hGRα-A, hGRα-B, hGRα-C1, hGRα-C2, hGRα-C3, hGRα-D1, hGRα-D2, and hGRα-D3), which are generated by alternative initiations of GRα mRNA translation [25]. These isoforms have progressively shorter N-terminal domains and different properties in terms of subcellular localization, expression, and function [26,27]. We speculated that the same molecular mechanisms may occur in the process of hGRβ mRNA translation into eight beta isoforms [15].

Within the target cell, glucocorticoids signal through their cognate receptor, which is located mainly in the cytoplasm, and forms a protein complex with immunophillins, such as FKBP51 and FKBP52, and heat shock proteins (e.g., HSP90 and HSP70) (Figure 1) [28,29]. Upon ligand binding, the structure of the receptor changes, thereby allowing the GR to dissociate from the interacting proteins and then move to the nucleus. In the nucleus, the ligand-activated hGRα forms dimers (homo-or heterodimers) that bind to specified DNA sequences, which are termed glucocorticoid response elements (GREs). GREs are located in the regulatory or promoter regions of glucocorticoid-responsive genes, thereby increasing or decreasing their expression [30]. In addition, the ligand-bound hGRα may influence the transcription rate of several non-target genes by interacting as a monomer with other transcription factors, including the nuclear factor-κB (NF-κB) [31], the activator protein-1 (AP-1) [32], and signal transducers and activators of transcription (STATs) [33]. In addition to genomic actions, glucocorticoids may induce some rapid effects. These “nongenomic” effects occur in cells that do not have a nucleus, and do not depend on transcription/translation processes [34,35]. The molecular mechanisms, which are responsible for the nongenomic glucocorticoid effects, remain under investigation; however, in vitro studies have uncovered the role of a membrane-anchored hGRα, which activates rapid kinase signaling pathways, including those of the mitogen-activated protein kinases (MAPK) or the phosphatidylinositol 3-kinase (PI_3_K) (Figure 1) [36]. Finally, the GR has also been detected in mitochondria, suggesting a functional role of the receptor in key mitochondrial functions, including apoptosis, production of energy, calcium homeostasis, thermogenesis, and response to stressors [37]. More than two decades ago, GREs were identified in the regulatory regions of the mitochondrial genome, termed as “D-loop”, indicating the GR as regulator of mitochondrial transcription [37,38]. In addition to direct binding of GR onto mitochondrial GREs, the expression of some mitochondrial genes is indirectly regulated by interactions between the nuclear GR and GRE, leading to the induction of mitochondrial RNA-processing enzymes and transcription factors, as well as important nuclear respiratory factors (Figure 1) [39,40].

In humans, the glucocorticoid signal transduction pathway exerts its functions in an inverted U—shaped activity-effect curve [8,40,41]. The optimal effect contributes to homeostasis or eustasis, and is achieved in the middle of glucocorticoid signaling activity. Suboptimal effects may be present on each side of the inverted U—shaped curve, and can cause insufficient adaptation. This state is termed allostasis or cacostasis. Indeed, the impaired activation of the glucocorticoid signal transduction may have short- or long-term detrimental effects for humans. These conditions may present with clinical manifestations of glucocorticoid resistance or glucocorticoid hypersensitivity [40,41].

## 2. Primary Generalized Glucocorticoid Resistance or Chrousos Syndrome

Primary generalized glucocorticoid resistance, or Chrousos syndrome, is an extremely rare endocrine disorder that affects all tissues expressing the hGR, and is characterized by a generalized, partial decreased sensitivity to glucocorticoids [40,41,42,43,44,45,46,47,48,49,50,51]. Chrousos syndrome can be inherited in an autosomal recessive or dominant fashion or may be sporadic due to de novo genetic defects (point mutations, deletions or insertions) in the *NR3C1* gene [50,51]. In patients with Chrousos syndrome, the presence of defective hGRs in both hypothalamus and pituitary results in impaired glucocorticoid negative feedback loops, causing compensatory hypersecretion of corticotropin-releasing hormone (CRH), arginine vasopressin (AVP), and adrenocorticotropic hormone (ACTH) (Figure 2). The resultant high plasma ACTH concentrations lead to adrenal hyperplasia; stimulation of the adrenal cortex; activation of biosynthesis; and the release of cortisol, adrenal androgens (androstenedione, dehydroepiandrosterone (DHEA), and DHEA-sulfate (DHEAS)), and steroids with mineralocorticoid activity (corticosterone and deoxycorticosterone) (Figure 2) [50,51].

According to the pathophysiologic mechanisms discussed above, patients with Chrousos syndrome might present with biochemical hypercortisolism without manifestations of Cushing’s syndrome [51]. The clinical phenotype of the condition ranges from asymptomatic cases to clinical manifestations of mineralocorticoid excess (hypertension and hypokalemic alkalosis) and/or androgen excess, including ambiguous genitalia at birth with 46, XX, hirsutism, acne, male-pattern hair loss, precocious puberty, amenorrhea, hypofertility and menstrual irregularities in females, and oligospermia in males [40,41,42,43,44,45,46,47,48,49,50,51]. The increased secretion of CRH and AVP may cause anxiety and depression, while hypercortisolism may explain the chronic fatigue in these patients [51]. Glucocorticoid deficiency has been described in rare cases: in adult patients with chronic fatigue [47,52,53]; in a 2-year-old girl with generalized seizures, hypoglycemia, hypokalemia, increased arterial pressure and premature pubarche [54]; and in a newborn with hypoglycemia, severe hypertension, easy “fatigability” with feeding and growth hormone deficiency [55]. The molecular basis of Chrousos syndrome has been attributed to inherited or de novo genetic defects in the *NR3C1* gene, which are depicted in Figure 3 and presented in Table 1 [52,54,55,56,57,58,59,60,61,62,63,64,65,66,67,68,69,70,71,72,73,74,75,76,77,78,79,80,81,82,83,84,85,86].

The diagnostic workup of Chrousos syndrome includes a complete medical history with particular emphasis on symptoms and signs suggestive of impaired activity of the HPA axis or possible CNS dysfunction (e.g., visual impairment, headaches or seizures) [41,42,43,44,49]. In adolescent girls and women, any irregularity in menstrual cycles should be documented. The growth velocity and puberty progress in children and adolescents should be carefully assessed through clinical examination. In addition, specific attention should be given to any of the above-discussed clinical manifestations indicative of mineralocorticoid or/and androgen excess [41,42,43,44,49]. The initial endocrinologic evaluation includes the determination of the morning (08:00 h) concentrations of serum cortisol, plasma ACTH, serum aldosterone, plasma renin activity (recumbent and upright), androgens (androstenedione, testosterone, DHEA and DHEAS), and insulin. Triglycerides, total cholesterol, LDL, HDL, and fasting glucose concentrations should be simultaneously measured at 08:00 h. The confirmation of diagnosis of Chrousos syndrome depends on the determination of the 24 h urinary free cortisol (UFC) excretion on a minimum of two consecutive days. The 24 h UFC excretion might be increased by 50-fold in comparison with the upper normal values, while serum cortisol concentrations might reach 7-fold higher in comparison with the highest values of its normal range. Moreover, ACTH concentrations might be normal or elevated, whereas the circadian rhythms of the HPA axis remain within the normal range [41,42,43,44,49].

To evaluate how the HPA axis responds to exogenous synthetic glucocorticoids, patients with suspected Chrousos syndrome should undergo a dexamethasone suppression test. According to the standard protocol, progressively increasing doses of the synthetic glucocorticoid analogue dexamethasone (0.3, 0.6, 1.0, 1.5, 2.0, 2.5, and 3.0 mg) should be given per os to the patients at midnight every other day [41,42,43,44,49]. Serum cortisol concentrations are measured at 08.00 h the following morning. Crucially, serum dexamethasone concentrations should also be measured to eliminate the possibility of increased clearance or reduced absorption of this synthetic glucocorticoid analogue. Due to the impaired negative feedback loops, patients with Chrousos syndrome display resistance of the HPA axis following a dexamethasone suppression test. In addition to endocrinologic evaluation, a pituitary magnetic resonance imaging scan should be performed to exclude the presence of pituitary adenoma [41,42,43,44,49].

Thymidine incorporation assays and dexamethasone-binding assays performed on mononuclear cells of systemic circulation remain the appropriate ex vivo methods to confirm the diagnosis of Chrousos syndrome [41,42,43,44,49]. Compared to control participants, affected subjects show resistance to the suppression of phytohemaglutinin-stimulated thymidine incorporation following exposure to dexamethasone, and display lower binding of the administered tritiated dexamethasone to their mutant receptors. Finally, the identification of the genetic defects in patients with Chrousos syndrome is achieved by the Sanger sequencing of the *NR3C1* gene, including the junctions of introns and exons [41,42,43,44,49].

Mineralocorticoid-sparing synthetic glucocorticoids, such as dexamethasone, should be administered per os at high doses only in symptomatic patients with Chrousos syndrome (1–3 mg given once daily at night) [41,42,43,44,49]. The main aim of treatment in these patients is to effectively suppress the increased secretion of ACTH, thereby decreasing the concentrations of adrenal-derived steroids with mineralocorticoid or androgenic activity. To avoid the development of an ACTH-secreting adenoma, the dose of dexamethasone in affected patients should be strictly titrated, based on the clinical manifestations and biochemical findings of the patients [41,42,43,44,49].

## 3. Recent Advances in Primary Generalized Glucocorticoid Resistance or Chrousos Syndrome

In 2018, Vitellius et al. provided, for the first time, an estimation of the prevalence of *NR3C1* mutations in 100 patients who had bilateral adrenal hyperplasia, increased arterial pressure, and/or hypercortisolism, but without any stigmata of Cushing’s syndrome [79,87]. Notably, they found that five of these patients (5%) carried novel heterozygous *NR3C1* mutations. The two mutations (R469X and R477S) were identified in the DNA-binding domain, and the rest of them (R491X, Q501H, and L672P) were identified in the ligand-binding domain of hGRα [79,87]. This high prevalence indicates that a number of cases with *NR3C1* genetic defects might be present and not identified. Consequently, clinicians should recommend Sanger sequencing of the *NR3C1* in patients with symptoms and signs suggestive of Chrousos syndrome [87].

Tatsi and coworkers recently described an interesting case of a female patient who initially presented at almost 3 years old with hypertension, hypoglycemic seizures, and hypokalemia [68]. The patient had endocrinologic findings suggestive of Chrousos syndrome, and was subsequently treated with antihypertensive medications, potassium supplementation, and dexamethasone at a dose of 1 mg daily. The case was first published by Nader et al., who identified a novel heterozygous point *NR3C1* mutation (c.2141G > A, p.R714Q) by using mRNA/cDNA, and proceeded to the functional characterization of this point mutation [54]. At the age of 11 years 9 months, the patient was receiving a 12 mg dose of dexamethasone daily, and was found to have several neuroimaging findings consistent with severe hypertensive encephalopathy. The daily dose of dexamethasone was increased to 14 mg to achieve better control of arterial hypertension [68]. Using whole-exome sequencing, Tatsi et al. identified two nonsynonymous heterozygous mutations in the *NR3C1* gene: the one previously published by Nader et al. [54], and a novel mutation in exon 2, c.592G > T, p.E198X. To the best of our knowledge, this is the first published case of two heterozygous point mutations being harbored in the *NR3C1* gene, suggesting that it is better to use whole-exome sequencing when searching for *NR3C1* genetic defects [68]. 

In addition to the significant contribution of next-generation sequencing to the identification of novel *NR3C1* mutations in patients with Chrousos syndrome, the last decade has seen the tremendous progress of structural biology in an attempt to determine how a specific genetic defect of the *NR3C1* gene leads to such conformational change(s) of the hGRα, ultimately leading to tissue resistance to glucocorticoids [72,74,75,76,88,89]. In addition to our previously published work on the structural biology of hGRαV423A [74], hGRαV575G [75], hGRαH726R [76], and hGRαT556I [72], two recent publications shed more light on the structural alterations caused by already known *NR3C1* mutations that resulted in the clinical manifestations of Chrousos syndrome [88,89]. Monteiro et al. performed computational analysis for eight mutations located in the ligand-binding domain of the hGRα to provide a deeper understanding on the atom scale of how they cause glucocorticoid resistance [88]. They found that these mutations caused decreased binding to cortisol, as well as alterations in a number of loop conformations due to changes in the motion of the mutant protein [88]. In a second recently published study, Kaziales et al. used in vitro and in silico methods to describe the structural biology of the mutant hGRαL773P receptor [89]. The authors demonstrated significant conformational alterations regarding the binding of the mutant receptor to dexamethasone, to an NCOA-2 co-activator peptide, to DNA, and to HSP90 [89].

## 4. Primary Generalized Glucocorticoid Hypersensitivity Syndrome

Primary generalized glucocorticoid hypersensitivity is also extremely rare, and represents the inverse condition of primary generalized glucocorticoid resistance syndrome. This syndrome is characterized by increased tissue glucocorticoid sensitivity; therefore, affected patients usually present with the cardinal clinical manifestations of metabolic syndrome, such as obesity, dyslipidemia, insulin resistance, and hypertension. However, the concentrations of serum cortisol are low or even undetectable due to compensatory hypoactivation of the HPA axis (Figure 2) [40,42,43,44]. The molecular basis of this condition remains under investigation. A small number of cases with primary generalized glucocorticoid hypersensitivity syndrome have been associated with *NR3C1* polymorphisms, including the N363S and *Bcl1*, or mutations [40,42,43,44,90]. 

The first case of primary generalized glucocorticoid hypersensitivity syndrome was a woman who harbored an activating *NR3C1* mutation and presented with visceral obesity, type 2 diabetes, increased arterial pressure, and dyslipidemia [90]. Sequencing analysis of the *NR3C1* gene showed a heterozygous point substitution G→C at nucleotide position 1201, which caused a substitution of aspartic acid (D) for histidine (H) at amino acid position 401, which is located in the N-terminal domain of hGRα. Charmandari et al. performed in vitro studies to investigate the function of hGRαD401H [90]. Transactivation studies showed that hGRαD401H induced the expression of glucocorticoid target genes in response to increasing concentrations of dexamethasone. However, the mutation D401H did not alter the affinity of the mutant receptor for the ligand, had no effect on the time required to complete its cytoplasmic to nucleus translocation, did not affect its ability to bind to GREs of target genes, and did not change its ability to interact with the GRIP1 coactivator [90]. Notably, a number of cases with primary generalized glucocorticoid hypersensitivity syndrome and no mutated *NR3C1* gene have also been reported [91,92,93,94,95]. 

Our research team has also described a novel case of transient primary generalized glucocorticoid hypersensitivity syndrome, and performed transcriptomics during the acute and the resolution phase of the disease [96]. The case index was a 9-year-old girl with a long-standing history of weight gain, moon facies, buffalo hump, acanthosis nigricans, purple skin striae, myopathy, hirsutism, and gradual deceleration of the growth velocity. Remarkably, the 08:00 h plasma ACTH and serum cortisol concentrations were undetectable (<1 pg/mL and 0.025 mcg/dL, respectively), and both remained at low concentrations during the 24 h period. A CRH stimulation test was performed and revealed no increase in the concentrations of plasma ACTH and serum cortisol. A pituitary MRI scan was normal [96]. Both the clinical manifestations and endocrinologic findings were suggestive of generalized glucocorticoid hypersensitivity. Sequencing analysis of the *NR3C1* gene did not identify any activating mutations or polymorphisms associated with glucocorticoid hypersensitivity. Notably, the clinical features of the patient were gradually eliminated over the next 3 months, indicating the diagnosis of transient generalized glucocorticoid hypersensitivity [96]. RNA was isolated from the peripheral blood mononuclear cells at the acute phase and the resolution phase of the disease, and RNA samples were subjected to transcriptomic analysis using RNA sequencing. Of note, 106 genes were induced, and 797 were repressed during the acute phase, as compared with the resolution phase. Most of them were identified as NF-κB responsive genes, suggesting that a viral or bacterial protein might function as a hGRα co-activator, leading to increased transcriptional activity of the receptor [96].

## 5. Novel Insights in Primary Generalized Glucocorticoid Hypersensitivity Syndrome

In addition to the D401H mutation and the well-studied polymorphisms of the *NR3C1* gene, a novel hyperactive G1376T single-nucleotide polymorphism (SNP) isoform was recently identified in hospitalized patients with burn injuries on at least 20% of their total body surface area [97]. This SNP resulted in a single substitution of glutamine to valine at amino acid position 459, which is located in the DNA-binding domain of the receptor. The hGRαG459V-mediated transcriptional activity was investigated in reporter assays following exposure to hydrocortisone, the hGR antagonist RU486 (mifepristone) alone, or RU486 followed by hydrocortisone [97]. Upon exposure to hydrocortisone, the hGRαG459V displayed 30-fold greater transcriptional activity compared to the wild-type hGRα. Surprisingly, the hGRαG459V showed elevated transcriptional activity when exposed to RU486. When transfected cells were treated with RU486 followed by hydrocortisone, the transcriptional activity of the mutant receptor was repressed, but greater than that noted upon exposure only to RU486 [97]. These unique transcriptional properties of the mutant receptor hGRαG459V might be attributed to the localization of this substitution. Further studies are needed to investigate the function of the mutant receptor [97].

From a therapeutic point of view, Liu et al. reported a case of a 27-year-old male with clinical manifestations and endocrinologic findings suggestive of primary generalized glucocorticoid hypersensitivity [98]. A pituitary MRI scan was normal, and an adrenal computed tomography (CT) scan revealed bilateral adrenal atrophy and an increased content of subcutaneous fat in the abdomen. The patient was treated using mifepristone with beneficial effects on electrolyte balance, lipid metabolism, and bone turnover [98]. Larger studies should confirm the therapeutic effects of mifepristone in patients with this rare condition.

## 6. Beyond *NR3C1* Gene Mutations and Polymorphisms: Tissue Glucocorticoid Sensitivity in the –OMICs Era

Primary generalized glucocorticoid resistance and hypersensitivity syndromes represent the two sides of the inverted U-shaped curve of the homeostatic function of glucocorticoid signaling. Undoubtedly, these allostatic conditions may have detrimental adverse effects for humans [8]. However, significant variability in tissue glucocorticoid sensitivity has been documented among healthy adults, as evidenced by differences in therapeutic response and in the frequency of glucocorticoid adverse effects. In an attempt to identify a motif that could distinguish healthy individuals with differences in tissue glucocorticoid sensitivity, we recruited 101 adults with unremarkable medical history [99,100]. The participants were given 0.25 mg of dexamethasone at midnight, and serum cortisol concentrations were measured the following morning at 08:00 h. According to their serum cortisol concentrations, participants were rank-ordered into the 10% most glucocorticoid sensitive and the 10% most glucocorticoid resistant. One month later, DNA, RNA, and plasma samples were collected from the 22 subjects. A sequencing analysis of the *NR3C1* gene did not identify any mutations or polymorphisms in the 22 selected subjects [99,100]. In addition, no statistically significant difference was found in telomere length between the glucocorticoid-resistant and the glucocorticoid sensitive group [99]. A transcriptomic analysis through RNA sequencing revealed a large percentage of differentially expressed genes, which play an important role in the maintenance of telomere length, systemic lupus erythematosus, Alzheimer’s disease, and Parkinson’s disease [99]. Untargeted metabolomics analysis in plasma samples from the selected participants showed a detrimental profile of metabolites in the most glucocorticoid-sensitive compared to the most glucocorticoid-resistant of the healthy subjects, and offered a metabolic motif that could distinguish between the two groups of participants in subsequent larger studies [100]. We speculate that the observed differences among healthy individuals at the –OMICS level might be attributed to epigenetic modifications that occurred during acute or chronic stress at early life stages. Nevertheless, tissue glucocorticoid sensitivity, in the context of primary generalized glucocorticoid resistance, hypersensitivity syndromes, or healthy individuals, remains at the forefront of translational research successfully connecting clinical observations with laboratory findings.

## Figures and Tables

**Figure 1 ijms-22-10839-f001:**
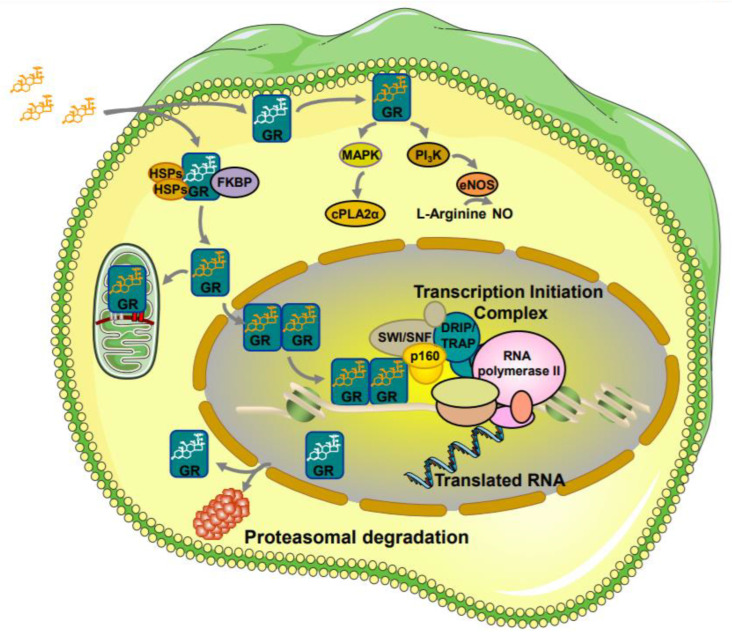
Signaling pathways of genomic, nongenomic and mitochondrial glucocorticoid actions. cPLA2α: cytosolic phospholipase A2 alpha; DRIP/TRAP: vitamin D receptor-interacting protein/thyroid hormone receptor-associated protein complex; eNOS: endothelial nitric oxide synthetase; FKBP: immunophilins; GR: glucocorticoid receptor; HSP: heat shock proteins; MAPK: mitogen-activated protein kinases; NO: nitric oxide; p160: nuclear receptor coactivators p160; PI_3_K: phosphatidylinositol 3-kinase; SWI/SNF: switching/sucrose non-fermenting complex (modified from ref. [6]).

**Figure 2 ijms-22-10839-f002:**
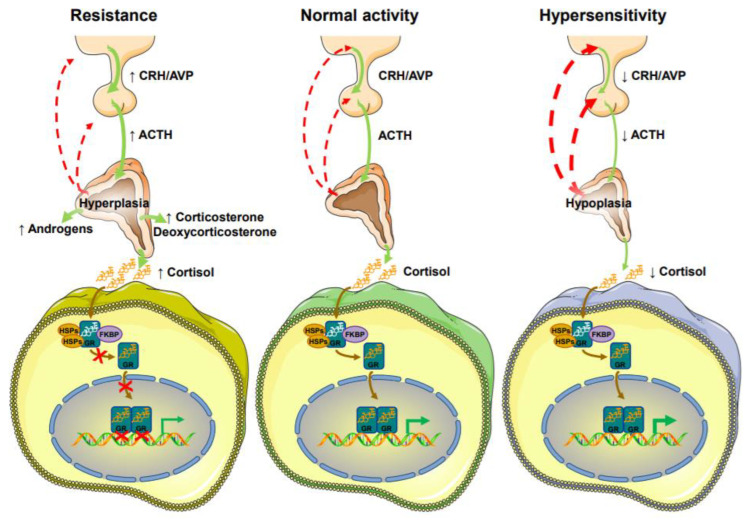
Pathophysiologic mechanisms of primary generalized glucocorticoid resistance and hypersensitivity syndromes. ACTH: adrenocorticotropic hormone; AVP: arginine-vasopressin; CRH: corticotropin-releasing hormone; FKBP: immunophilins; GR: glucocorticoid receptor; HSP: heat shock proteins.

**Figure 3 ijms-22-10839-f003:**
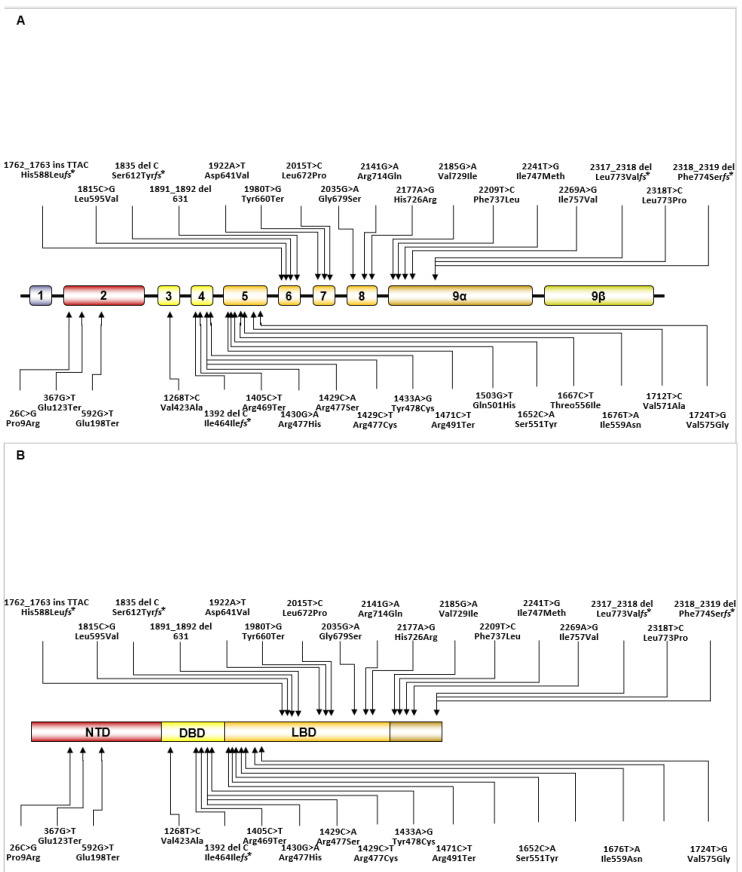
Schematic representation of the published genetic defects causing Chrousos syndrome in the *NR3C1* gene (**A**) and in the linearized hGR protein (**B**) (modified with permission from ref. [13], 2020, Nicolaides, N.C.; Chrousos, G.; Kino, T.).

**Table 1 ijms-22-10839-t001:** Mutations of the *NR3C1* Gene Causing Chrousos Syndrome: Molecular Mechanisms and Clinical Manifestations.

**Reference**	**cDNA**	**Amino Acid**	**Molecular Mechanisms**	**Genotype**	**Phenotype**
Chrousos et al. 1982 [52] Hurley et al. 1991 [56] Charmandari et al. 2004 [57]	1922 (A→T)	641 (D→V)	Transactivation ↓ Affinity for ligand ↓ (x3) Nuclear translocation: 22 min Abnormal interaction with GRIP1	Homozygous	Hypertension Hypokalemic alkalosis
Karl *et al*. 1993 [58]	4 bp deletion in exon-intron 6		hGRα number: 50% of control Inactivation of the affected allele	Heterozygous	Hirsutism Male-pattern hair-loss Menstrual irregularities
Malchoff et al. 1993 [59] Charmandari et al. 2004 [57]	2185 (G→A)	729 (V→I)	Transactivation ↓ Affinity for ligand ↓ (x2) Nuclear translocation: 120 min Abnormal interaction with GRIP1	Homozygous	Precocious puberty Hyperandrogenism
Karl et al. 1996 [60] Kino et al. 2001 [61] Charmandari et al. 2004 [57]	1676 (T→A)	559 (I→N)	Transactivation ↓ Decrease in hGR binding sites Transdominance (+) Nuclear translocation: 180 min Abnormal interaction with GRIP1	Heterozygous	Hypertension Oligospermia Infertility
Ruiz et al. 2001 [62] Charmandari et al. 2006 [63]	1430 (G→A)	477 (R→H)	Transactivation ↓ No DNA binding Nuclear translocation: 20 min	Heterozygous	Hirsutism Fatigue Hypertension
Ruiz et al. 2001 [62] Charmandari et al. 2006 [63]	2035 (G→A)	679 (G→S)	Transactivation ↓ Affinity for ligand ↓ (x2) Nuclear translocation: 30 min Abnormal interaction with GRIP1	Heterozygous	Hirsutism Fatigue Hypertension
Mendonca et al. 2002 [64] Charmandari et al. 2004 [57]	1712 (T→C)	571 (V→A)	Transactivation ↓ Affinity for ligand ↓ (x6) Nuclear translocation: 25 min Abnormal interaction with GRIP1	Homozygous	Ambiguous genitalia Hypertension Hypokalemia Hyperandrogenism
Vottero et al. 2002 [65] Charmandari et al. 2004 [57]	2241 (T→G)	747 (I→M)	Transactivation ↓ Transdominance (+) Affinity for ligand ↓ (x2) Nuclear translocation ↓ Abnormal interaction with GRIP1	Heterozygous	Cystic acne Hirsutism Oligo-amenorrhea
Charmandari et al. 2005 [66]	2318 (T→C)	773 (L→P)	Transactivation ↓ Transdominance (+) Affinity for ligand ↓ (x2.6) Nuclear translocation: 30 min Abnormal interaction with GRIP1	Heterozygous	Fatigue Anxiety Acne Hirsutism Hypertension
Charmandari et al. 2007 [67]	2209 (T→C)	737 (F→L)	Transactivation ↓ Transdominance (+) Affinity for ligand ↓ (x 1.5) Nuclear translocation: 180 min	Heterozygous	Hypertension Hypokalemia
Nader et al. 2010 [54] Tatsi et al. 2019 [68]	2141 (G→A)	714 (R→Q)	Transactivation ↓ Transdominance (+) Affinity for ligand ↓ (x2) Nuclear translocation ↓ Abnormal interaction with GRIP1	Heterozygous	Hypoglycemia Hypokalemia Hypertension Mild clitoromegaly Advanced bone age Precocious pubarche
McMahon et al. 2010 [55]	2 bp deletion at nt 2318-9	773	Transactivation ↓ Affinity for ligand: absent No suppression of IL-6	Homozygous	Hypoglycemia Fatigability with feeding Hypertension
Bouligand et al. 2010 [69]	1405 (C→T)	469 (R→X)	Transactivation ↓ Ligand-binding sites ↓ No DNA binding No nuclear translocation	Heterozygous	Adrenal hyperplasia Hypertension Hypokalemia
Trebble et al. 2010 [70]	1835delC	Ser612Tyr*fs**	Protein expression: (ࢤ) Ligand binding: (ࢤ) Cytoplasm to nuclear translocation: (ࢤ) Dominant negative effect: Yes Transactivation: (ࢤ)	Heterozygous	Fatigue
Zhu Hui-juan *et al*. 2011 [71] Nicolaides et al. 2016 [72]	1667 (G→T)	556 (T→I)	Transactivation ↓ Transrepression ↑ Affinity for ligand ↓ Nuclear translocation ↓ Abnormal interaction with GRIP1	Heterozygous	Adrenal incidentaloma
Donner et al. 2012 [73]	2317_2318delCT	Leu773Val*fs**	Protein expression: slightly reduced Transactivation: Decreased Dominant negative effect: No Ligand binding: (ࢤ)	Heterozygous	Hypertension
Roberts et al. 2013 [74]	1268 (T→C)	423 (V→A)	Transactivation ↓ Affinity for ligand: Normal No DNA binding Nuclear translocation: 35 min Interaction with GRIP1: Normal	Heterozygous	Fatigue Anxiety Hypertension
Nicolaides et al. 2014 [75]	1724 (T→G)	575 (V→G)	Transactivation ↓ Transrepression ↑ Affinity for ligand ↓ (x2) Nuclear translocation ↓ Abnormal interaction with GRIP1	Heterozygous	Melanoma Asymptomatic daughters
Nicolaides et al. 2015 [76]	2177 (A→G)	726 (H→R)	Transactivation ↓ Transrepression ↓ Affinity for ligand ↓ (x2) Nuclear translocation ↓ Abnormal interaction with GRIP1	Heterozygous	Hirsutism Acne Alopecia Anxiety Fatigue Irregular menstrual cycles
Velayos et al. 2016 [77]	1429 (C→T)	477 (R→C)	Not studied yet	Heterozygous	Mild hirsutism Asymptomatic mother
Velayos et al. 2016 [77]	1762_1763insTTAC	588 (H→L*5)	Not studied yet	Heterozygous	Hirsutism Anxiety Chronic fatigue
Vitellius et al. 2016 [78]	1429 (C→A)	477 (R→S)	No Transactivation Affinity for ligand: Normal No DNA binding Nuclear translocation ↓	Heterozygous	Adrenal incidentaloma
Vitellius et al. 2016 [78]	1433 (A→G)	478 (Y→C)	Transactivation ↓ Affinity for ligand: Normal DNA binding ↓ Nuclear translocation ↓	Heterozygous	Adrenal incidentaloma
Vitellius et al. 2016 [78]	2015 (T→C)	672 (L→P)	No Transactivation No Affinity for ligand No DNA binding No Nuclear translocation	Heterozygous	Adrenal incidentaloma
Vitellius et al. 2018 [79]	1471C>T	Arg491Ter	Transactivation: (ࢤ)	Heterozygous	Bilateral adrenal hyperplasia
Vitellius et al. 2018 [79]	1503G>T	Gln501His	Transactivation: Decreased	Heterozygous	Bilateral adrenal hyperplasia
Al Argan et al. 2018 [80]	1392del	465 (I→S)	Not studied yet	Heterozygous	Chronic fatigue Acne Hirsutism Hypomenorrhea
Cannavò et al. 2019 [81]	1915C>G	Leu595Val	Not performed	Not available	Hirsutism Amenorrhea Hypertension
Molnár et al. 2018 [82]	2141G→A	714 (R→Q)	Studied in Nader et al. 2010	Heterozygous	Infertility Asymptomatic sister
Vitellius et al. 2019 [83]	1980T>G	Tyr660Ter	Transactivation: (ࢤ)	Heterozygous	Hypertension
Tatsi et al. 2019 [68]	592G>T	Glu198Ter	Not performed	Compound heterozygous	Hypertensive encephalopathy
Lin et al. 2019 [84]	26C>G	Pro9Arg	Not performed	Heterozygous	Hypertension
Ma et al. 2020 [85]	1652C>A	Ser551Tyr	Ligand binding: Decreased Cytoplasm to nuclear translocation: Decreased Transactivation: Decreased	Homozygous	Fatigue Hypokalemia Hypertension Polyuria
Paragliola et al. 2020 [86]	367G>T	Glu123Ter	Not performed	Heterozygous	Chronic fatigue Anxiety Hirsutism Irregular menstrual cycles Infertility

↓: decreased; +: present; (x2): two-fold increased; ↑: increased; (ࢤ): absent.

## Data Availability

Data sharing not applicable.

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
