# Peer review of "Primary Generalized Glucocorticoid Resistance and Hypersensitivity Syndromes: A 2021 Update"

_ijms, 2021, doi:10.3390/ijms221910839_

Round 1
Reviewer 1 Report
In their current manuscript entitled, "Primary generalized glucocorticoid resistance and sensitivity syndromes: 2021 Update," Nicolaides et. al., do a very nice job summarizing the data on both genomic and non-genomic glucocorticoid signaling as well as current understanding of glucocorticoid resistance and hypersensitivity in patients, including information on the various mutations that have been associated with human disease.
All and all, the manuscript is very well written, easy to understand, and covers an interesting topic. The one caveat is that the figures did not download with copy of the manuscript so I was unable to check them for clarity and/or typos. Otherwise, I feel this manuscript can be published as is.
Author Response
We thank the Reviewer for his/her comments. We have now re-submitted a word file containing the manuscript and the figures.

Reviewer 2 Report
The manuscript seems to be interesting, but I would like to have the figures. Some English revision is necessary.
Author Response
We thank the Reviewer for these comments. We have now re-submitted a word file containing the manuscript and the figures. We have also improved the quality of the English language throughout the manuscript.

Round 2
Reviewer 2 Report
It is a fantastic job done by the authors, the manuscript was improved, and the figures help understand some concepts. In my opinion, the manuscript is suitable for publication.